# Magnetically-driven phase transformation strengthening in high entropy alloys

Changning Niu[1], Carlyn R. LaRosa[1], Jiashi Miao[1], Michael J. Mills[1] & Maryam Ghazisaeidi[1]

CrCoNi alloy exhibits a remarkable combination of strength and plastic deformation, even superior to the CrMnFeCoNi high-entropy alloy. We connect the magnetic and mechanical properties of CrCoNi, via a magnetically tunable phase transformation. While both alloys crystallize as single-phase face-centered-cubic (fcc) solid solutions, we find a distinctly lower-energy phase in CrCoNi alloy with a hexagonal close-packed (hcp) structure. Comparing the magnetic configurations of CrCoNi with those of other equiatomic ternary derivatives of CrMnFeCoNi confirms that magnetically frustrated Mn eliminates the fcc-hcp energy difference. This highlights the unique combination of chemistry and magnetic properties in CrCoNi, leading to a fcc-hcp phase transformation that occurs only in this alloy, and is triggered by dislocation slip and interaction with internal boundaries. This phase transformation sets CrCoNi apart from the parent quinary, and its other equiatomic ternary derivatives, and provides a new way for increasing strength without compromising plastic deformation.

[1] Materials Science and Engineering, Ohio State University, 2041 College Rd, Columbus, OH 43210, USA. Correspondence and requests for materials should be addressed to M.G. (email: ghazisaeidi.1@osu.edu)

Simultaneous increase in strength and ease of plastic deformation (or ductility) is the ultimate goal for most structural materials, but is usually mutually exclusive[1–4]. A new class of metallic alloys, known as high-entropy alloys (HEA), have shown great promise in overcoming this conflict. HEAs are comprised of multiple metallic elements—usually five or more—in equal or near equal concentrations that crystallize as single-phase solid solutions with simple crystal structures[5,6]. In particular, one of the most-studied HEA systems to show the coveted combination of strength and ductility is the equiatomic CrMnFeCoNi with a single-phase fcc structure[7,8]. Later studies of various derivatives of this alloy showed that the ternary CrCoNi —also crystallizing in a single fcc phase—exhibits even higher cryogenic strength, ductility and fracture toughness, making it superior to its quinary parent HEA or any of its other equiatomic fcc ternary or quaternary derivatives[9,10]. Despite current interest in CrCoNi, the exact origin of its excellent mechanical properties still remains unclear. Networks of nanotwins have been observed in this alloy following extensive deformation, as well as the quinary version, and are thought to be responsible for the simultaneous increase in strength and ductility via accommodation of dislocations at the coherent boundaries[7,11,12]. However, the ease of twin formation alone cannot explain the drastic difference in behavior of this alloy compared to other twinning-deformation-dominated materials or the superiority of the ternary CrCoNi over the quinary CrMnFeCoNi.

Here we show that the origin of these extraordinary properties, in CrCoNi, is the existence of a nano-structured hcp phase, caused by metastability of the fcc phase at low temperatures. A remarkable aspect is that the hcp laths form only localized to stacking faults and twin interfaces, suggesting that an additional mode of strengthening in these alloys is operative. We then show that this mechanism is suppressed by magnetic frustration in CrMnFeCoNi and its other equiatomic ternary single-phase derivatives.

## Results

**FCC to HCP phase transformation.** Figure 1 shows an example of clearly developed hcp layers within a nanotwin in CrCoNi, compared to the frequently observed nanotwin structure in the CrMnFeCoNi alloy. These images are obtained by atomic resolution scanning transmission electron microscopy (STEM), in high-angle annular dark field (HAADF) mode. Samples are deformed to about 55% true strain and are imaged along ⟨110⟩ zone axis such that these nanotwin and hcp lath interfaces are being viewed edge-on. The "center of symmetry (COS)" visualizations in panels b and d indicate deviation from the fcc symmetry, and thus "hot" regions have local hcp stacking. COS maps can greatly facilitate visualization of stacking faults and twins by detecting deviation from FCC atomic arrangements[13], in this case along a ⟨101⟩ zone axis. Single hcp layers represent the A-B-A stacking at boundaries of nanotwins, as seen exclusively in the CrMnFeCoNi alloy (c) and (d), while thicker hcp stacking layers displaying regions that have transformed into hcp laths, are commonly observed in the ternary alloy under similar deformation conditions.

Next, we demonstrate the driving force for the above fcc-hcp phase transformation in CrCoNi alloy, using density functional theory (DFT) calculations. Figure 2 shows the relative energy of hcp and fcc phases of this alloy. DFT calculations on 6 permutations of a special quasi-random structure (SQS)[14] of the CrCoNi are performed. Considering the vibrational entropic contributions to the free energy—within the quasi-harmonic approximation—shows that the hcp phase is favored over the fcc phase at lower temperatures, particularly in the cryogenic regime

relevant to the experiments. Similarly, pure Co exhibits a hcp-fcc phase transformation at around 695 K while Ni is always stable in the fcc structure. As a benchmark, relative free energies of hcp and fcc phases for Co and Ni are also included in Fig. 2. The quasi-harmonic free energies overestimate the transition temperature of Co, but shows consistent favorability of fcc Ni over the considered temperature range.

In contrast to CrCoNi, hcp structures have only been reported in the five-element CrMnFeCoNi after the extreme loading conditions of diamond-anvil cell experiments[15]. In spite of extensive, multiscale investigation, we have observed only very rare and thin hcp laths in the quinary alloy after conventional deformation experiments. In addition, free energy calculations for the quinary alloy, such as the ones shown in Fig. 2, pose significant practical challenges. We have detailed these issues and included the consequently non conclusive estimates for relative free energy values of this alloy in Supplementary Fig. 1 along with additional discussions in Supplementary Note 1.

**Effect of magnetism.** Figure 3 shows the effect of magnetism on phase stability of these alloys. Figure 3a, b show the total energy of all fcc and hcp configurations of CrCoNi and CrMnFeCoNi respectively. To test the importance of magnetic effects, we performed both magnetic and nonmagnetic calculations by switching on and off spin polarization. In CrCoNi, the hcp phase is consistently favored over fcc, even when the magnetic effects are turned off. The average energy of either phase is reduced by considering magnetism, but the difference between the average energies of hcp and fcc phases does not change significantly. Also, none of the hcp configurations, spin polarized or not, are ever higher in energy than any of the fcc phases. The behavior of the CrMnFeCoNi alloy is more complicated; even though the average energy of the hcp phase is lower than that of the fcc phase, it is clear that the difference between the two average energies is reduced with magnetic contributions. Also, there are several occasions where the fcc and hcp energies overlap, suggesting that the relative stability of these phases depends on the local distribution of atoms in the alloy. The different levels of complexity in the magnetic structure of two alloys is evident when comparing the respective atomic magnetic moments.

Figure 3c, d show the atomic magnetic moments for all calculations in fcc and hcp phases of the ternary and the quinary alloy, respectively. In the ternary alloy, clear patterns emerge for Ni and Co. Cr magnetic moment is mostly negative in the hcp phase but switches to positive values in some of the fcc configurations. The magnetic moments on Ni and Co in the quinary alloy show similar trends as in the ternary alloy, while Cr moments show a wider range of fluctuations. More importantly, Fe and Mn have more complicated trends; Mn moment switches between positive and negative values in both phases, and Fe appears to experience negative magnetic moments in the hcp structure.

Pure Mn is an antiferromagnetic element. When combined with ferromagnetic elements Ni, Co and Fe and another antiferromagnetic element Cr, preferred antiparallel alignment of Mn spins cannot be completely satisfied. Pure Mn in fcc structure would be magnetically frustrated due to geometric constraints on nearest neighbor exchange interactions[16]. In case of the HEA, the frustration also depends on composition, type and concentration of other elements. It is expected that frustration can be engineered by changing Mn concentration. This is in fact consistent with recent results of Li et al.[17] that show reducing Mn content favors the hcp phase in nonequiatomic subsets of the quinary HEA. Note that Cr magnetic moments are frustrated as well, however, they have smaller values and their

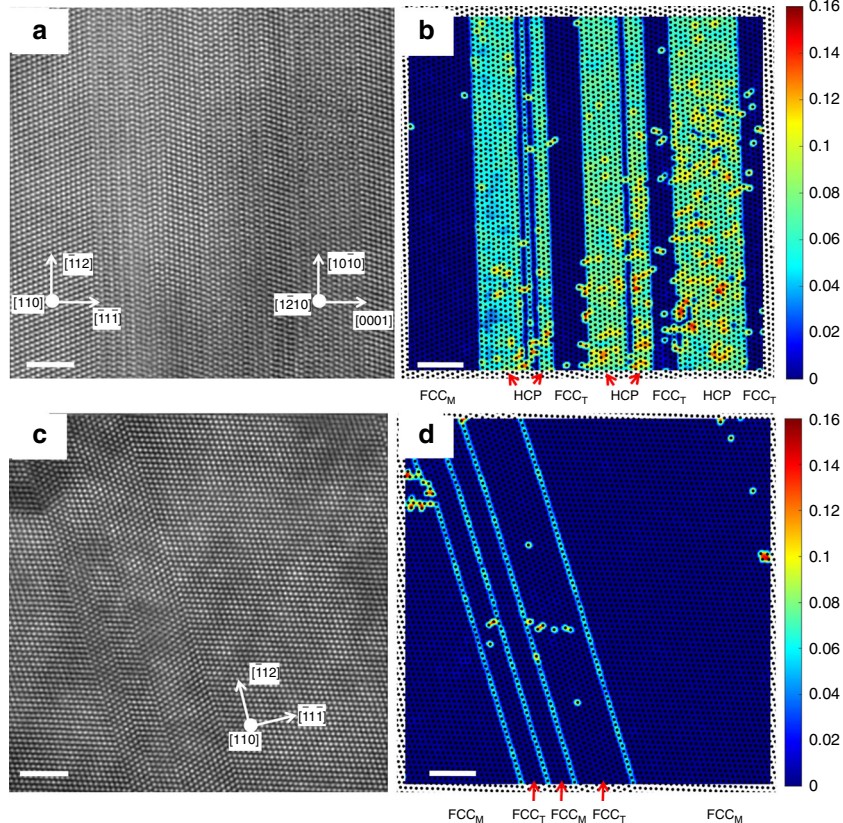

**Fig. 1** Deformation substructure in CrCoNi and CrMnFeCoNi alloys tested at cryogenic temperature. HAADF-STEM images show (**a**) a well-developed nanotwin-HCP lamellar structure in CrCoNi and (**c**) a nanotwin structure in CrMnFeCoNi. The corresponding center of symmetry maps are shown in **b**, **d**, respectively. The center of symmetry maps highlight the degree to which the local environment deviates from a FCC crystal (cf. Methods section). $FCC_M$ and $FCC_T$ refer to the matrix and twin in the FCC crystal, respectively. All scale bars represent 2 nm

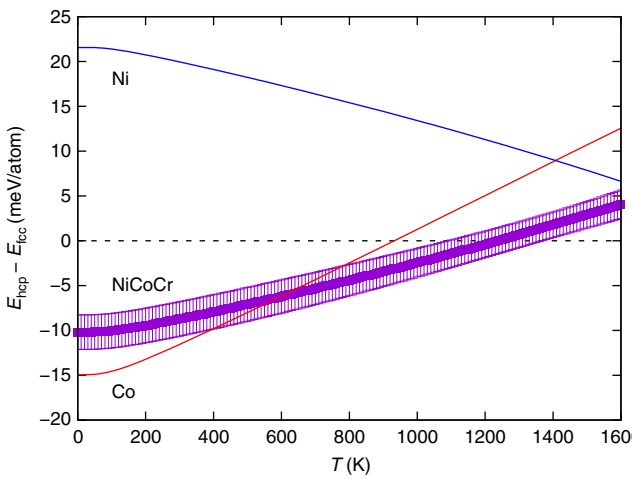

**Fig. 2** Free energy difference between hcp and fcc phases as a function of temperature for Ni, Co and CrCoNi. The calculations are performed with DFT using quasi-harmonic approximation. The error bars for CrCoNi are standard errors of $\bar{E}_{hcp} - \bar{E}_{fcc}$, difference in average energies of each phase, calculated over six SQS cells with different random distributions

contribution is not enough to counterbalance the chemical preference for hcp in the ternary alloy.

Furthermore, in order to probe the role of individual elements in more detail, we repeated the above calculations for other ternary equiatomic derivatives of the quinary HEA. According to[8,9], FeCoNi, MnCoNi and MnFeNi are the only combinations,

in addition to CrCoNi, that crystalize in a single-phase fcc solid solution and are thus studied here. Figure 4 presents the fcc and hcp energies as well as atomic magnetic moments for each of these alloys. Important conclusions can be drawn from this figure as follows. First, note the difference in average fcc and hcp energies $\Delta E = E_{fcc} - E_{hcp}$, with and without considering magnetism. This comparison reveals the magnetic contribution to the relative phase stability. Note that $\Delta E^{magnetic}$ and $\Delta E^{non\ magnetic}$ are almost the same in CrCoNi. On the other hand, while $\Delta E^{non\ magnetic}$ is approximately 10 and 15 meV in MnCoNi and MnFeNi respectively, $\Delta E^{magnetic}$ is practically zero in both alloys. In case of FeCoNi, the magnitude of $\Delta E^{magnetic}$ is slightly larger than that of $\Delta E^{non\ magnetic}$. Therefore, in alloys containing Mn, magnetism has the strongest effect on the relative stability of fcc and hcp phases. Second, comparing FeCoNi to MnCoNi or MnFeNi reveals a unique feature regarding magnetic structure of alloys containing Mn. In case of FeCoNi, the spins of all ferromagnetic elements are aligned perfectly and there is no scatter in magnetic moment distributions of element types. As a result there is a clear ground state which is the fcc phase in this case. Substituting either Fe or Co with Mn atoms eliminates the energy difference between the hcp and fcc phases, as noted above. In addition, the magnetic moments of Mn atoms are clearly frustrated, in the sense that a unique magnetic order for the alloy cannot be reached; Mn atoms can have either parallel or antiparallel alignments with respect to their neighbors. This observation shows the coexistence of magnetic frustration and comparable fcc and hcp energies. Therefore, no driving force for fcc to hcp phase transformation exists in any of these alloys. Finally, comparing CrCoNi and MnCoNi differentiates the behavior of Cr and Mn, which are

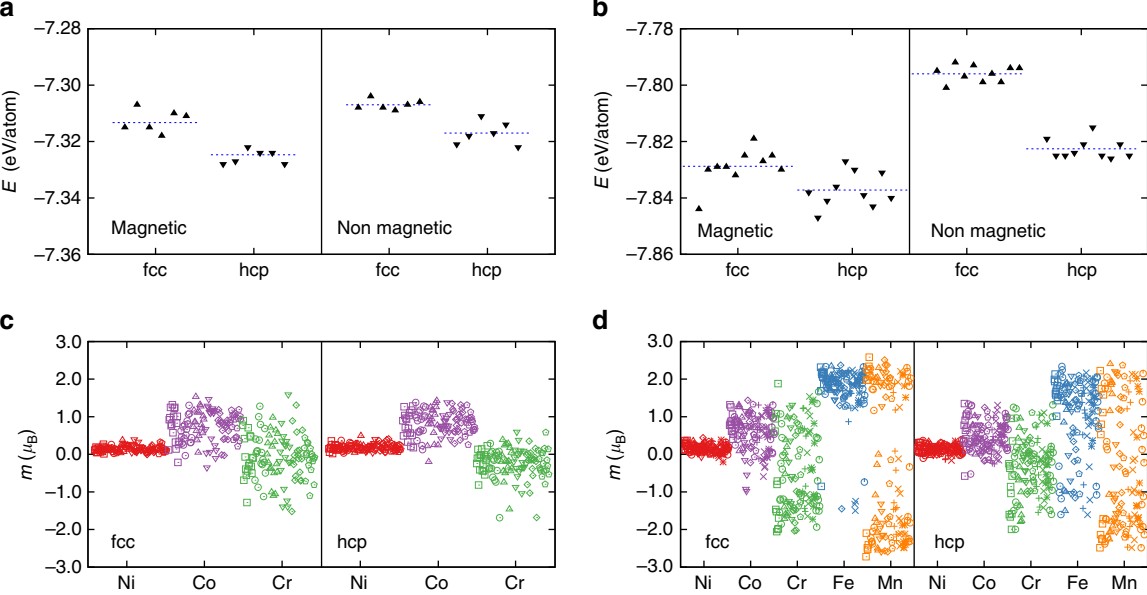

**Fig. 3** Magnetic effects on the energies of hcp and fcc phases in CrCoNi and CrMnFeCoNi. **a**, **b** The total energy of fcc and hcp phases for the two alloys with and without spin polarization. Different random distributions of each composition are achieved by permuting atom types in a fcc or hcp SQS cell, resulting in 6 and 10 permutations for the ternary and quinary alloys respectively. Average energies are indicated by the dashed lines. In CrCoNi, the hcp phase is always favored with respect to fcc, even without spin polarized calculations. **c**, **d** The magnetic moments per atom for all calculations in the ternary and quinary alloy, respectively. The quinary alloy experiences a strong magnetic frustration, that results in an overlap between hcp and fcc energies

both antiferromagnetic in pure elemental form. Magnetic frustration also exists in the fcc CrCoNi, as Cr magnetic moments span a range of positive and negative values. This scatter decreases in the hcp phase, as most of Cr magnetic moments are negative. Overall, Cr magnetic moments seem to be strongly affected by the magnetic moments of other elements in their vicinity. On the other hand, Mn moments follow a rather bimodal distribution, fluctuating between approximately 3 and $-3\,\mu_B$ regardless of their environment. This comparison provides empirical evidence for the different natures of magnetism in Cr and Mn. Moreover, comparison between CrCoNi and MnCoNi alloys, with varying concentrations of Cr and Mn, elucidates the different magnetic behavior of Cr and Mn further and is included in the Supplementary Note 2 and Supplementary Fig. 2.

**Dislocation mechanisms for fcc to hcp phase transformation.** The above analysis highlights the unique combination of chemical and magnetic properties in CrCoNi, which distinguishes this alloy from its parent quinary and other ternaries of the same family; a lower energy of the hcp phase compared to the fcc phase, indicates a driving force for the fcc-hcp phase transformation in response to shear deformations that would bring the closed-packed planes from an ABCABC type stacking in fcc to ABABAB stacking in hcp. This phase transformation can be achieved by glide of $1/6\langle112\rangle$-type Shockley partial dislocations.

Figure 5 shows the energy pathways corresponding to glide of Shockley partials, with the same Burgers vector, on adjacent (111) planes. Glide of the first Shockley partial creates an intrinsic stacking fault (isf). Subsequent glide of second and third partials on neighboring (111) planes, creates an extrinsic stacking fault and a 3-layer twinned region, separated by two twin boundaries, respectively. Path 1 in Fig. 5 shows the energetics of this process from DFT calculations. This type of calculation has been used before to establish the generalized planar fault energies in fcc systems[18]. To compare the energetics of twin formation with those of the hcp phase transformation, we also considered two

additional paths once the isf is formed. In path 2, the second Shockley partial is placed on the second neighboring (111) plane, skipping one plane in between. This forms 4 layers of atoms, with hcp coordination, from the isf as shown in Fig. 5a. The third path compares the energetics of twin growth with that of hcp formation, from an already formed twin boundary. The formation of the hcp phase commences with the motion of a Shockley partial on the plane adjacent to the twin boundary, and can proceed with subsequent partials passing on every other (111) plane. Figure 5b shows the average fault energies over 36 configurations with various local chemistry at the fault plane. The following features are notable: First, the isf value is negative on average. This is expected, because atoms creating an isf have hcp coordinations locally and the hcp phase has lower energy than fcc. Second, even though the isf energy value is negative, creation of this isf requires a significant positive energy barrier. This explains why the fcc structure does not spontaneously transform into the hcp phase. Third, the hcp formation, whether from an isf or a twin boundary, lowers the energy and requires a lower barrier to overcome compared to the competing processes. Negative stacking fault energies have been reported for CrCoNi[10] before, but previous analysis stops at connecting the low stacking fault energies to ease of twin formation. This is true in any material with low, but not necessarily negative, fault energies. Our results confirm that hcp formation is in fact favored to twin formation in CrCoNi and are consistent with direct HAADF-STEM observations such as the ones in Fig. 1.

Note that the behavior described above corresponds to the average response of the alloy. Fluctuations in local chemistry result in a scattering in fault energy values. In order to study the effect of local chemistry on the fault energies, we considered all 12 layers in our supercell and sheared them in all three $1/6\langle112\rangle$-type directions, as explained in the methods section. Figure 5c shows the distribution of energy values for all calculations. As shown in this figure, even though the average isf value is negative, positive values are also possible depending on the arrangements of different atom types. Therefore, the fact that

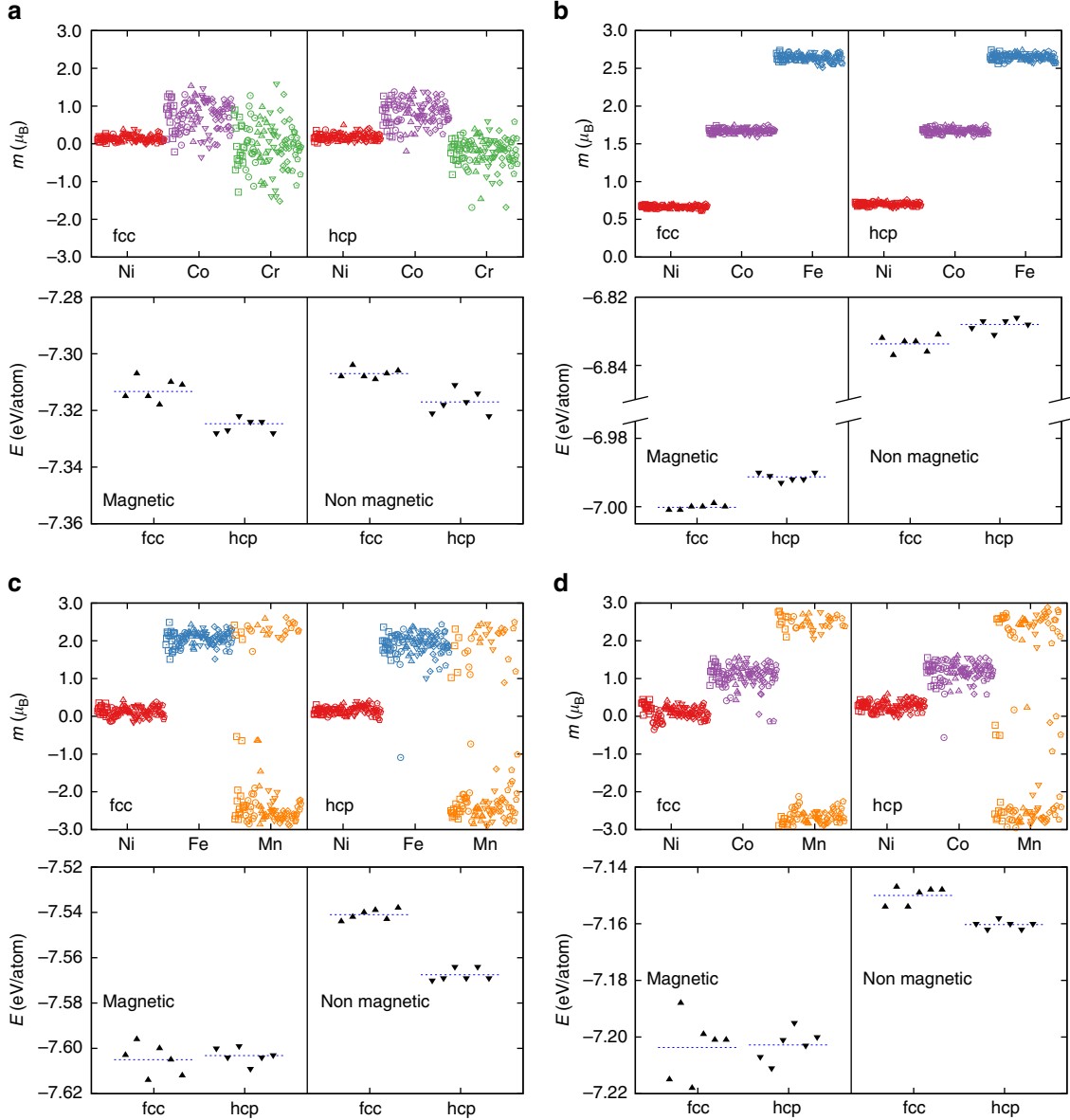

**Fig. 4** Magnetic effect on hcp-fcc energies in equiatomic ternary derivatives of CrMnFeCoNi. **a** corresponds to CrCoNi which is repeated from Fig. 3 for comparison. Panels **b**, **c** and **d** correspond to FeCoNi, MnFeNi and MnCoNi respectively. In each panel the first row shows the atomic magnetic moments and the second row compares the fcc and hcp energies of 6 configurations of each alloy, computed with DFT with and without considering magnetism. As evident in **c**, **d** only alloys containing Mn have similar energies in both fcc and hcp phases accompanied by significant frustration of Mn magnetic moments

on average hcp regions can form readily in CrCoNi does not imply that there are no twin boundaries in this alloy.

In addition, Fig. 6a compares the stacking fault energy pathways in the CrMnFeCoNi alloy with those in the ternary CrCoNi alloy. The average behavior of both alloys follow similar trends, however, the deviation from average behavior is much more significant in the quinary alloy. Note the numerous overlaps between energies of the stable and unstable structures in the quinary alloy. Nonmagnetic calculations, in Fig. 6b, show much smaller scatter in energies, although variations are still expected due to the changing chemical environment. The overlap between stable and unstable nonmagnetic energy values is eliminated, with consistently favorable hcp energies. This is in line with the analysis presented in Fig. 3. We do not expect a significant difference between magnetic and nonmagnetic stacking fault energies in the ternary CrCoNi alloy, since the magnetic effect on hcp and fcc energies in this alloy is negligible according to Fig. 3a.

Finally, we present a mechanistic picture of hcp formation from dislocation reactions and interactions with other defects. Atomistic modeling of dislocation/defects interactions requires large supercells, beyond the scope of standard DFT methods and are best captured by classical potential calculations using reliable interatomic potentials. In the absence of such potentials for Ni-Co-Cr system, we use fcc Co as a surrogate system with negative stacking fault energy and a more favorable hcp phase at $T = 0\,K$ (cf. Fig. 2 and Supplementary Note 3 and Supplementary Table 1). In addition, the energy pathway calculations of pure Co and Ni, in the Supplementary Fig. 3, verify that fcc Co conforms to the average response of CrCoNi. Supplementary Fig. 4 compares the response of preexisting isf and twin boundaries in Ni and Co to applied shear. While shearing the isf and twin boundary in Ni results in formation of esf and twin growth respectively, both of these incidents lead to formation of an hcp layer in fcc Co. This confirms the predictions made by energy pathway calculations.

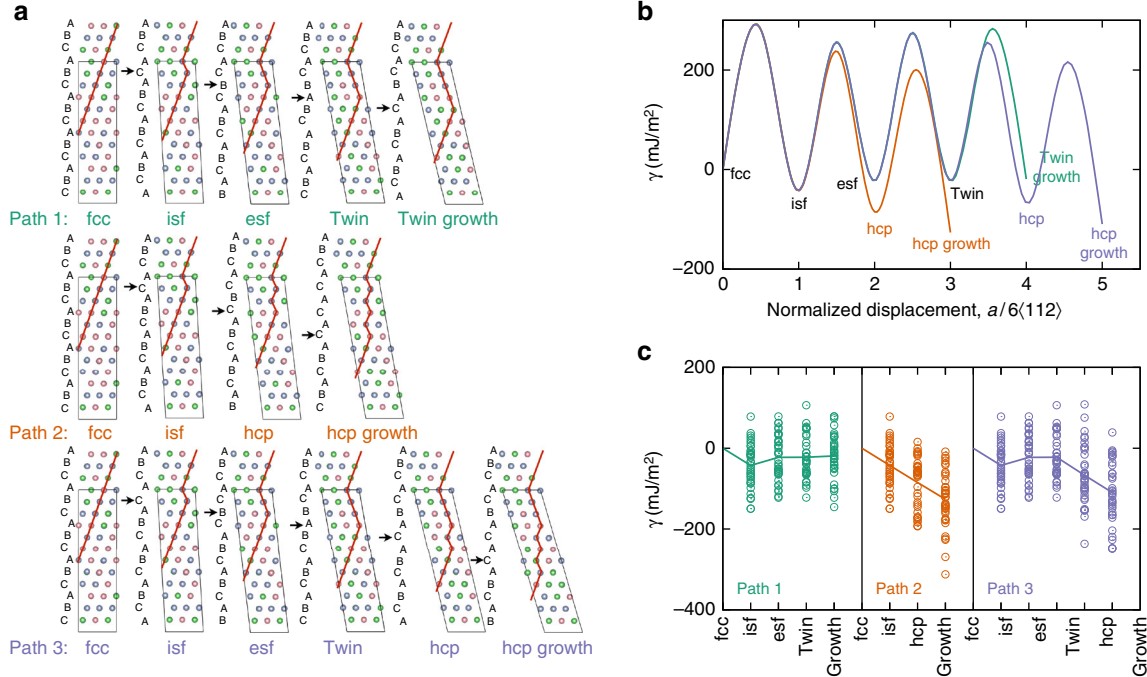

**Fig. 5** Ab initio generalized planar fault energies versus normalized shear displacements on successive {111} planes in CrCoNi. **a** The supercells used to calculate the energetics of three paths: Path 1 twin formation and growth, Path 2: hcp formation, and subsequent growth, from an intrinsic stacking fault and Path 3: hcp formation, and subsequent growth, from a twin boundary. Starting from the fcc structure, one-layer intrinsic fault (isf), two-layer extrinsic fault (esf) and three-layer twins are created by introducing Shockely partials with $b = 1/6\langle112\rangle$ on adjacent {111} planes. The hcp stacking is obtained by skipping one plane and introducing the next partial on the second neighboring plane. Black arrows indicate the successive planes where Schockley partials are introduced. **b** The fault energies, averaged over 36 configurations with different local chemistry at the fault plane. **c** The corresponding energies of all 36 configurations with average values marked by solid lines

Thus, fcc Co is a reasonable surrogate for the CrCoNi alloy, especially since local chemistry does not change the overall favorability of the hcp over fcc phase in this alloy, as demonstrated in Fig. 3b, and similar average behavior is capable of capturing the trends qualitatively. Further evidence, in favor of this choice, is provided in Fig. 7 where a HAADF observation of a dislocation/boundary interaction in CrCoNi is reproduced closely by atomistic simulation of fcc Co.

In addition to the homogeneous nucleation of the hcp layer, as described above, heterogeneous mechanisms—involving dislocation interactions—are also found. For example, Supplementary Fig. 5 shows formation of hcp layers from interaction of a 60° mixed dislocation with an existing twin boundary. The dislocation easily dissociates into 90° and 30° partial dislocations with an extended stacking fault, on the inclined glide plane that is impinging on the twin boundary. The extended stacking fault results from the negative fault energy, which eliminates any penalty for dissociation of partials. Under sufficient applied strain, both partials can be forced to move towards the twin boundary. The leading partial dislocation—whether the 90° or the 30° Shockley partial—is trapped at the twin boundary and subsequently acts as a source of stress concentration, facilitating the nucleation of new partial dislocations on the twin boundary. As a result, the original coherent twin boundary transforms into a 3-layer hcp structure. This hcp layer can further grow under additional resolved shear strain. Interaction of screw dislocations with the twin boundary results in twin growth, similar to what has been reported in other fcc systems[19–21]. Interaction of lattice dislocations, with already formed hcp layers, reveals important additional mechanisms that could promote increased strength without loss of ability to plastically deform. Supplementary Fig. 6, and Supplementary Movies 1–4, show the interaction of screw

and mixed dislocations with a well-formed hcp layer. In all cases, the leading partial is stopped at the fcc/hcp boundary requiring additional applied load to pass through the boundary. Under sufficient load, the screw dislocation penetrates the hcp layer, glides on the hcp prism planes and transfers to the fcc region on the other side. In case of the mixed dislocation, the trapped leading partials—whether it is the 90° or the 30° partial—again act as stress concentration sites and facilitate the nucleation of new partial dislocations, gliding on the hcp basal planes and transforming the hcp layer back to fcc stacking. Therefore, in all cases, the absorption of the leading partial by the boundary is accompanied by further plastic deformation, either through complete transfer of the dislocation or creation of additional mobile dislocations, parallel to the twin plane.

The observed hcp laths are akin to the ε phase in TRIP (transformation-induced plasticity) steels. This ε-martensite phase forms during low temperature deformation of the high Mn austenitic steels, which also exhibit extraordinary work hardening and ductility[22]. However, in TRIP steels, a relatively large volume fraction of the austenite transforms to the ε-martensite phase. In contrast, our observations in the CrCoNi alloys suggest a distinctly different behavior: the alloy remains predominantly fcc and the hcp laths form only localized to stacking faults and twin interfaces. In this process, twin boundaries pose a barrier to dislocations shearing across the interface, while simultaneously redirecting the shear parallel to the interface, thereby forming hcp laths at the twin boundary. In addition, the hcp layers can form by glide of extended dislocations on adjacent planes, which is facilitated by the negative average stacking fault energy of the alloy. The coherent interface between hcp and fcc phases also stops and redirects the incoming dislocations, providing an additional strengthening

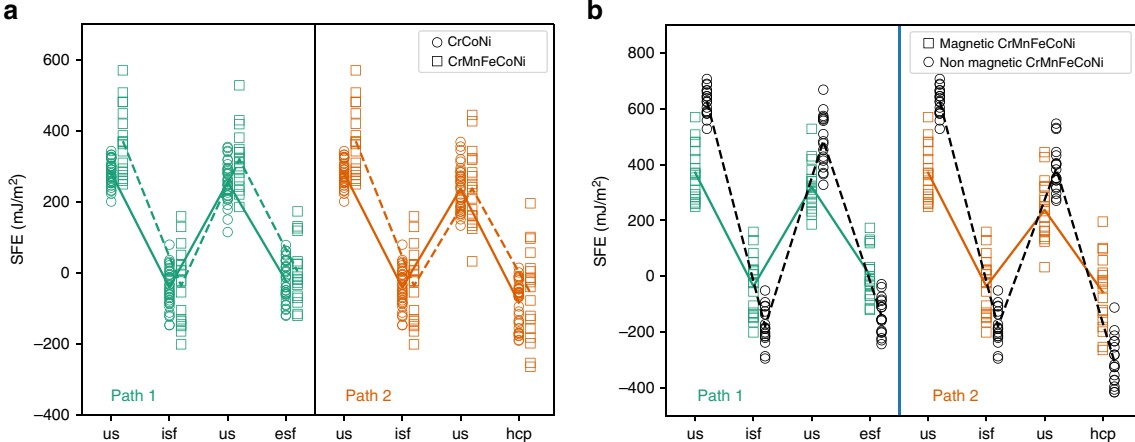

**Fig. 6** Planar fault energies in CrMnFeCoNi. **a** The fault energies with those of the ternary CrCoNi. The values in the ternary alloy are the same as those presented in path and are repeated here for easy comparison. Paths 1 and 2 correspond to the formation of extrinsic stacking faults versus hcp laths, as defined in Fig. 5. **b** The effect of magnetism on fault energies in the quinary alloy. The average behavior of both alloys follow similar trends, however, the scatter relative to the average behavior is much more significant in the quinary alloy. Note several overlaps between energies of the stable and unstable structures in the quinary alloy. Nonmagnetic calculations show much smaller scatter in energies, although variations are still expected due to the changing chemical environment. The overlap between stable and unstable nonmagnetic energy values is eliminated, with consistently favorable hcp energies

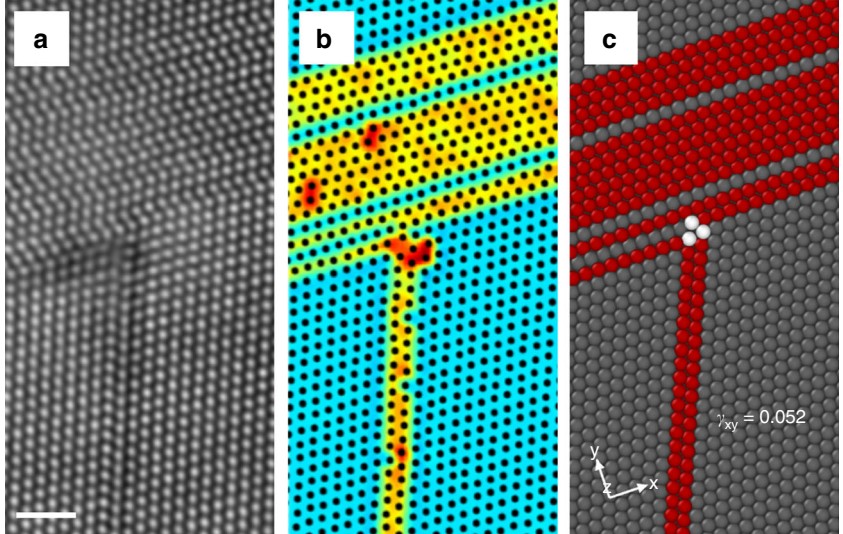

**Fig. 7** Comparison between experimental observation of deformation mechanisms in CrCoNi and atomistic simulation of fcc Co as a surrogate system. **a** A HAADF-STEM image of what appears to be the interaction between incoming dislocations and the existing hcp lamella with the corresponding center of symmetry map, shown in **b**. **c** A similar structure created by the simulation of two screw dislocations interacting with an hcp region of the same size in fcc Co. The second screw dislocation is out of the frame of the image. The clear similarity between the experimental and simulation images suggests that the structure in **a**, **b** are likely to have been created from a similar dislocation interaction with already formed hcp regions. The scale bar in **a** represents 1 nm

source. Strong interface barriers such as large angle grain boundaries ordinarily would be expected to cause highly localized stress concentrations at the tips of dislocation pile-ups that could lead to premature crack initiation[23]. Conversely, while the twin/hcp lath structures provides a strong barrier to slip transmission, the redirection of dislocations along the boundary smears out the damaging stress concentrations. With increasing strain, unusually large lattice rotations (greater than 30°) are created within individual grains[24], which further support the process described.

The limited volume fraction of hcp phase, even after large plastic strain, suggests that the structure may heal itself back to the fcc structure. Indeed we have observed removal of hcp layers, healing the structure back to the fcc state has been observed directly in the simulations shown in Supplementary Fig. 6. Since

the twin boundaries that acted as the nuclei for the initial hcp layer formation are still present, we envision that the hcp formation process can happen again with subsequent dislocation interactions on the intersecting slip system. Thus, slip on the intersecting slip system is inhibited (strengthening), while the internal interfaces can absorb the oncoming dislocation content (recovery).

We have shown the existence of a nanostructure with hcp phase in the equiatomic CrCoNi—an alloy widely considered to have a single-phase fcc structure. We observed a delineation between CrCoNi—with hcp as the ground state independent of the local atomic arrangements—and the CrCoMnFeNi HEA, where the magnetic frustration resulting from the addition of Mn breaks a clear trend for relative hcp vs fcc energies. In addition,

we showed that there is no driving force for this phase transformation in any other equiatomic ternary derivatives of this HEA. These findings emphasize the fact that high-entropy alloys with more constituent elements are not necessarily superior to their derivatives. Instead, the combination of chemical and, as emphasized presently, magnetic identities of elements govern the properties. An important consequence of this finding is that new alloying combinations can be predicted based on similar favorable magnetic properties. These results provide a new paradigm for design of single-phase alloys with metastable structures through computational guidance and experimental validation.

## Methods

**Computations**. Density functional theory (DFT) calculations are performed with the Vienna ab initio Simulation Package (VASP)[25], using the projector augmented wave (PAW) method[26] within the generalized gradient approximation of the exchange-correlation functional as determined by Perdew, Burke, and Ernzerhof (GGA-PBE)[27]. The valence electron configurations for the bulk calculations are Ni:$3d^94s^1$, Co:$3d^84s^1$, Cr:$3d^54s^1$, Fe:$3d^74s^1$, Mn:$3d^64s^1$. Collinear spin polarization (ISPIN = 2) is enabled in all magnetic calculations. We have tested some configurations with non-collinear spin settings and found that all magnetic orders remain collinear. A cut-off energy of 350 eV is used, and the Methfessel-Paxton smearing method (ISMEAR = 1) is applied with SIGMA = 0.2 in all DFT calculations. These settings combined with the choice of k-mesh, given below for each model, guarantees total energy convergence within 1 meV/atom. Three sets of atomistic models are built for various calculations of phase stability and generalized stacking fault and hcp lath formation energy pathways. The first set includes two 6-atom supercells of fcc and hcp consisting of six (111) layers for the calculation of pure Ni and Co. The two supercells use gamma-centered k-mesh of $19 \times 19 \times 4$. These two 6-atom models are then expanded by $3 \times 3 \times 1$ for the phonon vibration calculations using the finite displacement method, which use gamma-centered k-mesh of $4 \times 4 \times 2$. The second set includes two 54-atom supercells of fcc and hcp consisting of six (111) layers for the calculation of CrCoNi phase stability. The two supercells use gamma-centered k-mesh of $4 \times 4 \times 2$. The same supercells are used for phonon vibration calculations, which have almost identical supercell shape and volume as the expanded pure models. Phonon vibrations are studied on the first and the second set of atomistic models via the finite displacement method[28,29] as implemented in the software Phonopy[30] with its VASP interface. Quasi-harmonic approximation (QHA) of phonon vibrations takes volume dependence of phonon properties into consideration, while harmonic approximation is assumed at each volume image. Ionic relaxation of these models uses a strict force criterion of 1 meV/Å. Electronic relaxation uses criterion of $10^{-8}$ eV for the finite displacement method. The QHA calculations use up to 9 volume images with a lattice parameter ranging from 98 to 106% near the equilibrium value. Phonon bands of all volume images are examined to confirm that there is no imaginary components in any phonon band structure. The last set includes a 108-atom supercell of fcc consisting of 12 (111) layers for the shear calculations of CrCoNi. Shears of $\frac{1}{6}a_0\langle11\bar{2}\rangle$ are applied in sequence to its layers to introduce stacking faults. This supercell and its derivative supercells use gamma-centered k-mesh of $4 \times 4 \times 1$. In the shearing calculations using the third set of atomic models, the original fcc supercell is fully relaxed with a free cell volume and free atomic positions, while the faulted supercells are relaxed with only free atomic positions (selective dynamics on the unstable structures and fully free on the metastable structures). All these shearing models use a criterion of $10^{-6}$ eV for electronic relaxation and a criterion of 10 meV/Å for ionic relaxation. In the present study, all alloy supercells are special quasi-random structures (SQS)[14] generated by the mcsqs code of the ATAT package[31]. Their correlation functions match perfectly with corresponding ideal random alloys for the first nearest neighbors.

**Experiments**. Details of alloy preparation, mechanical properties and microstructure characterization of CrCoNi alloy and CrMnFeCoNi alloy were reported in recent papers[24] and [32] respectively. TEM specimens of alloy were machined from tensile specimens at 45 degrees with respect to the loading axis using electrical discharge machining (EDM). These TEM specimens were mechanically polished to a thickness of approximately 100 µm, followed by electropolishing for perforation in an electrolyte consisting of 20% perchloric acid in methanol at −30 ℃ and a voltage of 10–13 V. TEM specimens of CrMnFeCoNi alloys were prepared using a FEI Helios focus ion beam system. All FIB lift-out specimens were extracted from necking regions of tensile specimens where largest plastic deformation is expected. Final thinning of FIB lift-out TEM samples was accomplished at a voltage of 5 keV to minimize damage. FIB TEM samples were further cleaned using a Fischione 1040 nanomill before observation. Atomic resolution STEM characterization was conducted on a probe-corrected and monochromated Titan3 80–300TM STEM in the high-angle annular dark field (HAADF) mode. A nonlinear drift distortion correction Matlab program[33] was used to reduce scan distortion and noise associated with atomic resolution HAADF-STEM images. Center of symmetry (COS) analysis[13] examines the centroids of the atomic columns, determined from HAADF-STEM images through cross-correlating the image with Gaussian peaks with defined maxima and provides the measure of the deviation of the determined atomic positions from a FCC crystal structure. The COS maps thus illustrates deformation substructure including stacking faults, nanotwins and deformation-induced HCP phase.

**Data availability**. All relevant data are available upon request from the authors.

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

## Acknowledgements

C.N., C.L. and M.G. were supported by the National Science Foundation Grant DMR-1553355. M.G. also acknowledges partial support through the Air Force Office of Scientific Research Grant FA9550-17-1-0168. J.M. and M.M. were supported by the National Science Foundation under the contract No. DMR-60050072. Computational resources were provided by the Ohio Supercomputer Center. We are grateful to Dr. Guillaume Laplanche of the Ruhr-Universität Bochum, Institut für Werkstoffe for kindly providing deformed specimens of the quinary alloy, and Dr. Hongbin Bei of the Oak Ridge National Laboratory for samples of the ternary alloy.

## Author contributions

M.G. and M.M. designed the research and analyzed the data. C.N. performed the DFT calculations. C.L. performed classical potential atomistic simulations. J.M. performed high-angle annular dark field imaging and analysis. M.G. wrote the draft with input from all authors. All authors discussed the results.

## Additional information

**Competing interests:** The authors declare no competing interests.

