## [Peer Review File · Nature Communications]

Reviewers' comments:

Reviewer #1 (Remarks to the Author):

The manuscript is well written, providing seemingly solid evidence supporting the main claims, seen from the perspective of an experimentalist. The topic is of quite some interest to people working with high-entropy alloys, which is a rapidly developing field.

The reviewer is in general supporting its publication in Nature Communications, judging from its importance, novelty and technical soundness.

A few comments for the consideration of the authors.

1. On Fig. 1, the authors gave no explanation anywhere on what the subscripts T and M mean. Twin and matrix? It needs to be given for clarity.

2. On Figure 2, they mentioned that the quasi-harmonic free energies overestimate the transition temperature of Co. Since the deviation is close to 200K, which is way too significant, the reviewer is wondering what this can mean to the accuracy of the transition temperature for NiCoCr, and its indication on the fcc-hcp phase transformation as reflected in Fig. 2. Also, why not also draw the energy difference for CoCrFeMnNi? It would give an interesting scenario: NiCoCr resembles Co, while CoCrFeMnNi resembles Ni. What can that indicate? Something beyond magnetic frustration, as is the focus in the current work? The reviewer is really keen to see comments from the authors on this.

3. The reviewer is not convinced on what is delivered from Fig. 3 (b) and (d). Comparing cases when the spin polarization is switched off and on, obviously the energy difference between the fcc and hcp phase changes dramatically. However, comparing the change of magnetic moments per atom, I would say the change for Fe, rather than Mn, is most significant between the fcc and hcp phase. Referring to their cited Nature paper on how reducing the Mn content can favor the hcp phase in non-equiatom quinary (should be quaternary instead) HEA, it is noted that the Fe content increases with the reduction of Mn. So, it's due to Mn, or rather due to Fe? How can they argue for it?

4. There is a statement on page 9 where they said twinning can sometimes win energetically against hcp formation for NiCoCr. That is really a concern. What does that indicate? If we compare that case with CoCrFeMnNi, does that mean when this (twinning wins over hcp formation) happens, the mechanical behavior of NiCoCr would resemble CoCrFeMnNi? So the mechanical performance of NiCoCr will not always be superior to CoCrFeMnNi? That is really confusing indeed. Are we solving the problem here, or creating more problems?

Reviewer #2 (Remarks to the Author):

The authors present a very interesting experimental result and an interesting theory to explain their result. I think that ultimately, the work is publishable in Nature Communications, but I have one reservation.

I am not quite certain that the connection between magnetic frustration and the observed results is made clear enough. I agree that the authors have established that spin-polarized GGA gives a different FCC-HCP energy difference than a non-spin polarized calculation. This, I think, points to the role of spin polarization in determining the energies of the competing structures. I also agree that the lattices in question will lead to frustration if there are strong nearest neighbor antiferromagnetic interactions. However, I cannot see in the analysis of the moments for the two alloys, an argument beyond the fact that since the moments on Mn are larger, the magnetic energy contributes a larger portion of the internal energy. The frustration part of the explanation escapes me, particularly since I expect magnetic frustration in both the HCP and FCC phases. Admittedly, the total energy change due to the inclusion of spin polarization is greater in the quinary than the ternary alloy, but I don't see how one attributes this directly to the magnetic frustration.

From my understanding, the experimental observation of interest is that deformation of the FCC phase in the ternary alloy leads to local transformation to the HCP phase after deformation. In the quinary alloy, however, the HCP phase is rarely seen after deformation. Though I have not digitized the data, it appears that the FCC-HCP energy difference for the ternary and quinary alloys is are within 2 meV/atom of one another - near the accuracy limit of the calculations.

Given this energy difference, is it possible that the primary difference between the two alloys is that the barrier to forming an ISF in the ternary phase is lower than in the quinary phase? (This is a kinetics issue, not a relative stability issue.) Also, if the frustration in the magnetization is the cause of the differing behaviors, can't this be demonstrated conclusively by examining the changes in magnetic properties (moments, energy contributions) associated with overcoming the barrier to formation of the ISF in both alloys? This analysis, to my mind, has the potential to establish the causality between the frustration and the experimental differences between the alloys that the authors claim.

There are also minor issues with details of the calculations that are not mentioned. How were the magnetization calculations performed? What magnetization flags were set in VASP? Also, what type and level of Fermi level smearing did the authors employ? It might be helpful to plot typical spin structures obtained from VASP for the two alloys in order that they can be compared.

I think that if the authors can carry out this analysis, or perhaps explain more clearly the connection between frustration and their experimental observations, the paper would be publishable in Nature Communications.

Reviewer #3 (Remarks to the Author):

The paper describes a new contribution to the strengthening of metals through a magnetic frustration mechanism. This is a new and surprising mechanism that should be of significant interest to a wide range of readers. The paper is reasonably well-written. The information is presented logically, and the figures convey essential information. Citation of earlier relevant work is good. Some attention to grammar is needed, especially with regard to the use of articles.

The paper claims that the absence of magnetic frustration in CoCrNi, relative to the five-component 'parent' alloy CoCrFeMnNi, produces a phase transformation from metastable fcc to the stable hcp phase upon the sequential passage of $1/6\langle 112 \rangle$ partial dislocations. Several scenarios are described in the manuscript to illustrate this localized phase transformation. However, three points require additional analysis and discussion. First, the slip-induced phase transformation involves the sequential passage of partial dislocations on every other $\langle 111 \rangle$ plane. The authors must explain why the system would choose to deform by the passage of partial dislocations on every OTHER $\langle 111 \rangle$ plane, rather than every $\langle 111 \rangle$ plane or $\langle 111 \rangle$ planes that are more widely separated. The authors clearly show that the total system energy is reduced for this situation AFTER the passage of the partial dislocations. However, how does the system 'know' this will be the result when a partial dislocation is first nucleated and before it propagates to form or grow the hcp phase? The possibility that the transformation occurs all at once, rather than growing progressively by the sequential nucleation and passage of many partial dislocations, also requires consideration and discussion.

Second, the authors must discuss the trailing partials, which are not mentioned at all in the manuscript. Once the trailing partial travels through the material, the stacking of $\langle 111 \rangle$ planes is returned to the original sequence and the hcp phase is destroyed. The separation between leading and trailing partials may be sufficiently small so that the growth of a thick hcp plate may be difficult. The authors must describe how the hcp phase may persist with respect to the movement of trailing partial dislocations.

Finally, the authors do a good job of demonstrating that the fcc to hcp transition can occur by the passage of leading partial dislocations on selected $\langle 111 \rangle$ planes, but they do not adequately describe how this localized phase transformation produces strengthening. This new strengthening mechanism is the top-line discovery of this paper, and so the authors must convince the readers that this phase transformation is responsible for the additional strengthening observed in CoCrNi relative to CoCrFeMnNi. Showing that the passage of partial dislocations in CoCrFeMnNi does not produce a localized fcc to hcp transformation (see the additional comment below) could provide circumstantial support for the strengthening of CoCrNi relative to CoCrFeMnNi, but it would be more convincing to describe the specific mechanism by which this localized phase transformation strengthens CoCrNi.

The authors demonstrate a localized fcc to hcp transformation associated with slip in CoCrNi by calculating the energy of the fcc and hcp phases (Fig. 3) and through simulations involving the sequential passage of partial dislocations (Fig. 4). Based on calculations of the energies of the fcc and hcp phases in CoCrFeMnNi in magnetic and non-magnetic states, the authors imply that slip does not induce this local phase transformation in CoCrFeMnNi. However, this is rather indirect proof of the absence of the localized fcc to hcp phase transformation, and it would be more convincing for the authors to perform simulations on CoCrFeMnNi as illustrated in Fig. 4 for CoCrNi. This is especially important since Fig. 3 shows that the hcp phase has a lower total energy than the fcc phase for some of the configurations in the magnetic state of CoCrFeMnNi (see Fig. 3b).

Three additional comments are provided to improve the quality, readability and accessibility of the work. First, Fig 1 will not be entirely clear to readers who are not expert electron microscopists. The term "center of symmetry" in the caption needs explanation in a way that's accessible to the non-specialist, and the FCCM and FCCT labels on Figs. 1b,d require definition. Second, the work describes the strengthening of CoCrNi relative to CoCrFeMnNi. However, this could equivalently be considered as weakening of CoCrFeMnNi relative to CoCrNi and simpler alloys (binary alloys and/or alloys best represented as dilute solutions of elements). The authors should discuss this perspective to better support the claim of a new "strengthening" mechanism in the concentrated CoCrNi alloy. Specifically, if this same mechanism is found in concentrated and/or dilute binary alloys, then the high entropy alloy in the present paper is actually weaker than more conventional alloys. This is a subtle but important point, since much of the HEA literature proposes that HEAs will be stronger than alloys with fewer elements. Finally, the order in which the elements are listed in the two alloys seems to be rather arbitrary (for example, NiCoCr rather than CoCrNi or CrCoNi). While there is no right or wrong order, this can make keyword searches less effective. The HEA field seems to be moving toward adoption of one of two standardized naming conventions – either listing the elements alphabetically or listing in order of increasing atomic number. It's recommended that the authors consider using one of these two conventions.

Reviewers' comments:

Reviewer #1 (Remarks to the Author):

Comment 0: “The manuscript is well written, providing seemingly solid evidence supporting the main claims, seen from the perspective of an experimentalist. The topic is of quite some interest to people working with high-entropy alloys, which is a rapidly developing field. The reviewer is in general supporting its publication in Nature Communications, judging from its importance, novelty and technical soundness. A few comments for the consideration of the authors.”

Response: We thank the reviewer for a thorough review of our manuscript.

Comment 1: “On Fig.1, the authors gave no explanation anywhere on what the subscripts T and M mean. Twin and matrix? It needs to be given for clarity.”

Response: We thank the reviewer for pointing this out. The T and M indeed refer to Twin and Matrix and are now clarified in Fig.1.

Comment 2: “On Figure 2, they mentioned that the quasi-harmonic free energies overestimate the transition temperature of Co. Since the deviation is close to 200K, which is way too significant, the reviewer is wondering what this can mean to the accuracy of the transition temperature for NiCoCr, and its indication on the fcc-hcp phase transformation as reflected in Fig.2. Also, why not also draw the energy difference for CoCrFeMnNi? It would give an interesting scenario: NiCoCr resembles Co, while CoCrFeMnNi resembles Ni. What can that indicate? Something beyond magnetic frustration, as is the focus in the current work? The reviewer is really keen to see comments from the authors on this.”

Response: Regarding the accuracy of the quasiharmonic method for prediction of the transition temperature in Co vs NiCoCr: The phase transition in Co is known to have a magnetic nature. We tested this by switching off magnetism for pure Co and found that, even at $T=0K$, the ground state would be fcc instead of hcp. However, in case of the ternary alloy, as we discuss in the subsequent parts of the paper, the favorability of hcp over fcc is independent of magnetic contributions: i.e hcp is still lower energy than fcc even with nonmagnetic calculations. Thus, since the phase transition of pure Co and CoCrNi alloy have different natures, we anticipate the quasiharmonic approximation to predict this temperature better for the alloy.

That said, we agree with the reviewer that the predicted transition T is not exact. However, the mechanisms proposed in this paper occur at cryogenic temperatures, far below the predicted T and it is thus safe to assume that the hcp

energy is lower than the fcc energy over the temperature ranges relevant to the experiments.

In addition, we agree with the reviewer's comments about the necessity to look into the quinary alloy in terms of its trend of hcp-fcc energetics with vibrational entropy included, however such calculations are practically very challenging to perform. We have detailed the challenges and some estimates to the free energies in the supplementary materials (c.f Fig S1 and pages 1 and 2 of the revised Supplementary materials). In summary, the main issue is achieving converged results upon performing finite displacement calculations to compute the force constants.

Nonetheless, in order to have an estimate for comparison, we attempted to calculate the relative hcp-fcc free energies of the quinary alloy, only within HA for 6 fcc structures and 10 hcp structures. In the end, we successfully collected the vibrational entropies for 5 fcc structures and 3 hcp structures, while the rest had electronic convergence issues. Corresponding average (hcp-fcc) energy difference with standard error of the mean values is plotted in the new Fig S1. Note that the error bars on this plot are significantly larger compared to those for the ternary alloy. Therefore, this data is not conclusive, but we feel it is important to include it to show the practical subtleties in performing accurate calculations on the quinary alloy. Given all the approximations, the general trend in the average behavior of the two alloys is similar but the deviation from average is significantly larger in the quinary alloy. This is consistent with the rest of our findings in this work.

We have added the following text to the main draft, to point out the changes made to the supplementary section.

Page 5, line 72: "In addition, free energy calculations for the quinary alloy, such as the ones shown in Figure 2, pose significant practical challenges. We have detailed these issues and included the consequently non conclusive estimates for relative free energy values of this alloy in the Supplementary materials."

Comment 3: "The reviewer is not convinced on what is delivered from Fig. 3 (b) and (d). Comparing cases when the spin polarization is switched off and on, obviously the energy difference between the fcc and hcp phase changes dramatically. However, comparing the change of magnetic moments per atom, I would say the change for Fe, rather than Mn, is most significant between the fcc and hcp phase. Referring to their cited Nature paper on how reducing the Mn content can favor the hcp phase in non-equiatom quinary (should be quaternary instead) HEA, it is noted that the Fe content increases with the reduction of Mn. So, it's due to Mn, or rather due to Fe? How can they argue for it?"

Response: First, switching magnetism on and off is meant to demonstrate the magnetic contribution to the phase energies. It is expected that the total energy of each phase should change drastically, but the effect on the relative energies is nontrivial. Also, we agree with the reviewer that it is hard to distinguish the role of individual elements in the quinary alloy. In order to provide more detail on the behavior of each element, we also analyzed all fcc equiatomic ternary combinations. This was originally in the supplementary materials, but we now include it as Fig.4 in the main text along with some additional discussion (Page 7, lines 111-145).

In summary, one can compare the difference between Fe and Mn, by comparing NiCoFe with NiCoMn. In case of NiCoFe, all magnetic moments are perfectly aligned in both fcc and hcp phases. There is no scatter in energy and in this case fcc is the clear ground state. Substituting Fe with Mn, eliminates the difference in fcc vs hcp energy and Mn moments are frustrated in both fcc and hcp phases (i.e a unique magnetic order cannot be achieved). The case of NiFeMn is also interesting, since Fe moments do not change significantly, while again Mn moments could align both parallel or antiparallel to those of other atoms. While this analysis does not answer the question of why fcc and hcp energies are similar, it clearly shows a trend for Mn.

The quinary alloy's magnetic distribution is much more complicated, and there seems to be a strong dependence of the particular atomic arrangement. Even Cr moments have a wider distribution than that in the ternary alloy, which shows the magnetic moment of a Cr atom depends strongly on its neighbors. The only conclusions we draw from this Fig.3 (b) and (d) are (1) a unique magnetic order cannot be reached, hence frustration and (2) the difference in fcc and hcp energies (and the driving force for the phase transformation) is diminished due to the magnetic effects, because there is a clear gap in these energies when magnetism is switched off.

Comment 4: “There is a statement on page 9 where they said twinning can sometimes win energetically against hcp formation for NiCoCr. That is really a concern. What does that indicate? If we compare that case with CoCrFeMnNi, does that mean when this (twinning wins over hcp formation) happens, the mechanical behavior of NiCoCr would resemble CoCrFeMnNi? So the mechanical performance of NiCoCr will not always be superior to CoCrFeMnNi? That is really confusing indeed. Are we solving the problem here, or creating more problems?”

Response: We appreciate the reviewer pointing out this potential source of confusion. The statement meant to distinguish between the “average” trends

versus local deviations from this trend. On average the hcp formation is more likely in NiCoCr compared to CoCrFeMnNi, hence the better properties. However, this does not mean that twin formation is not possible in NiCoCr at all. In fact, in the subsequent part, we present twin boundaries as a potential site for nucleation of hcp laths in this alloy. Moreover, Fig.5(c) shows the average behavior and the scatter in response depending on the local arrangement of atoms.

We have modified the referred sentence in page 11 line 185 as : “Therefore, the fact that on average hcp regions can form readily in CrCoNi does not imply that there are no twin boundaries in this alloy.”

Reviewer #2 (Remarks to the Author):

Comment 0: “The authors present a very interesting experimental result and an interesting theory to explain their result. I think that ultimately, the work is publishable in Nature Communications, but I have one reservation.”

Response: We thank the reviewer for a careful review and helpful suggestions.

Comment 1: “I am not quite certain that the connection between magnetic frustration and the observed results is made clear enough. I agree that the authors have established that spin-polarized GGA gives a different FCC-HCP energy difference than a non-spin polarized calculation. This, I think, points to the role of spin polarization in determining the energies of the competing structures. I also agree that the lattices in question will lead to frustration if there are strong nearest neighbor antiferromagnetic interactions. However, I cannot see in the analysis of the moments for the two alloys, an argument beyond the fact that since the moments on Mn are larger, the magnetic energy contributes a larger portion of the internal energy. The frustration part of the explanation escapes me, particularly since I expect magnetic frustration in both the HCP and FCC phases. Admittedly, the total energy change due to the inclusion of spin polarization is greater in the quinary than the ternary alloy, but I don’t see how one attributes this directly to the magnetic frustration.”

Response: The reviewer raises a couple of very important points. To help explain the connection between frustration and the fcc-hcp energy differences, we brought the analysis of ternary alloys, originally in the supplementary materials, to the main text and added some discussions. The magnetic structure in the quinary alloy is too complicated and it is not possible to dissect the role of individual element from that alloy only.

The new Fig. 4 now analyzes all the fcc solid solution ternary combinations. Three major conclusions follow:

(1) $E_{\text{fcc}} - E_{\text{hcp}}$ is affected by magnetism the most, when the alloy contains Mn. Compare this energy difference, computed with and without spin-polarized settings, in each alloy. The difference almost vanished in Mn-containing alloys. So, magnetic contribution is important and there is something special about the Mn magnetic properties.

(2) Comparing NiCoFe with NiCoMn, reveals important difference between Fe and Mn. In NiCoFe all, magnetic moments are aligned in a parallel fashion, and there is a clear magnetic order. At the same times, the scatter in energy values is very small and there is a clear ground state, which happens to be fcc in this case. Substituting Fe with Mn, almost eliminates the energy difference and increases the scatter in energy. On the other hand a unique magnetic order is not achieved, since Mn moments can align either parallel or antiparallel to the other elements, hence frustration. The magnitude of magnetic moments of Mn and Fe are about the same, so the fact that these moments are bigger compared to other elements is not enough to explain the difference in behavior. In other words, while there is a significant difference between energy of each phase with and without magnetism, the relative fcc-hcp follow very different trends. Similar arguments can be drawn, by comparing NiFeMn with NiCoFe and NiCoMn.

(3) Comparing NiCoCr with NiCoMn shows also the difference in magnetic nature of Cr vs Mn. It's true that the Cr magnetic moments are smaller, but that's not the only difference. Cr magnetic moments can have virtually any value between $-1.7\mu_B$ and $1.7\mu_B$ while in the hcp phase, most of Cr atoms have small negative values. Mn moments on the other hand show a bimodal distribution (either approximately $+3\mu_B$ or $-3\mu_B$) in both phases. This is evidence than Cr magnetic moments depend strongly on its environment. We have compared these two alloys with various concentrations of Cr and Mn. The different magnetic behavior of Cr and Mn and their effects on stability of hcp vs fcc phase is evident from this analysis, which is presented in the revised Supplementary Materials Fig. S2.

Admittedly, the above is "empirical" evidence for a correlation between existence of magnetic frustration and relative phase energies. It does not establish a proof for "causality". The only way to make quantitative claims about the state of frustration is through calculation of exchange parameters. In the simplest form, the values can be approximated through a Heisenberg Hamiltonian (Ising Model) approximation, which works best in case of non localized spins, i.e not metals. Nevertheless, we attempted setting up the Ising model Hamiltonian for the SQS cells used in this study, but the linear fits failed to provide reasonable values as expected.

We have added additional text to the main draft on Page 7, lines 111-145.

Comment 2: “From my understanding, the experimental observation of interest is that deformation of the FCC phase in the ternary alloy leads to local transformation to the HCP phase after deformation. In the quinary alloy, however, the HCP phase is rarely seen after deformation. Though I have not digitized the data, it appears that the FCC-HCP energy difference for the ternary and quinary alloys is are within 2 meV/atom of one another - near the accuracy limit of the calculations.

Given this energy difference, is it possible that the primary difference between the two alloys is that the barrier to forming an ISF in the ternary phase is lower than in the quinary phase? (This is a kinetics issue, not a relative stability issue.) Also, if the frustration in the magnetization is the cause of the differing behaviors, can't this be demonstrated conclusively by examining the changes in magnetic properties (moments, energy contributions) associated with overcoming the barrier to formation of the ISF in both alloys? This analysis, to my mind, has the potential to establish the causality between the frustration and the experimental differences between the alloys that the authors claim.”

Response: First, we would like to point out that the average FCC-HCP difference in the ternary alloy is almost 11 meV/atom with no overlap between the energy of two phases. The average FCC-HCP energy in the quinary alloy is 8 meV/atom with several overlapping FCC and HCP energies. Even though the average energy differences are very close, as the reviewer points out, the main difference between the two alloys is in the deviation from average. In the ternary alloy, local fluctuations are small and never reverse the favorability of the hcp phase, while the deviation from average energies due to local arrangement of atoms is significant in the quinary alloy and results in overlap between fcc and hcp energies. In other words, in the quinary alloy, whether or not hcp is more favorable depends on the local arrangement of atoms.

The barriers to formation of ISF or hcp layers are included in Fig.5 for the ternary alloy. As suggested by the reviewer, we performed similar calculations on the quinary alloy for three types of planar defects: the intrinsic stacking fault (isf), the extrinsic stacking fault (esf), and the first layer of hcp phase and the corresponding barriers. These results are now shown in Fig 6. The average behavior of both alloys follow similar trends, however, the deviation from average behavior is much more significant in the quinary alloy. This is consistent with our previous predictions.

We have added the following text, page 11, lines 188-198:

“In addition, Figure 6 (a) compares the stacking fault energy pathways in the

CrMnFeCoNi alloy with those in the ternary CrCoNi alloy. The average behavior of both alloys follow similar trends, however, the deviation from average behavior is much more significant in the quinary alloy. Note the numerous overlaps between energies of the stable and unstable structures in the quinary alloy. Nonmagnetic calculations, in Figure 6 (b), show much smaller scatter in energies, although variations are still expected due to the changing chemical environment. The overlap between stable and unstable nonmagnetic energy values is eliminated, with consistently favorable hcp energies. This is in line with the analysis presented in Figure 3. We do not expect a significant difference between magnetic and nonmagnetic stacking fault energies in the ternary CrCoNi alloy, since the magnetic effect on hcp and fcc energies in this alloy is negligible according to Figure 3 (a).”

Comment 3: “There are also minor issues with details of the calculations that are not mentioned. How were the magnetization calculations performed? What magnetization flags were set in VASP? Also, what type and level of Fermi level smearing did the authors employ? It might be helpful to plot typical spin structures obtained from VASP for the two alloys in order that they can be compared.”

Response: The above details have been added to the methods section. For spin structures, we have now several figures (Figures 3, 5 and S2) with distribution of atomic magnetic moments in different alloys.

Comment 4: “I think that if the authors can carry out this analysis, or perhaps explain more clearly the connection between frustration and their experimental observations, the paper would be publishable in Nature Communications.”

Response: We thank the reviewer for pointing out the ambiguities in our presentation.

Reviewer #3 (Remarks to the Author):

Comment 0: “The paper describes a new contribution to the strengthening of metals through a magnetic frustration mechanism. This is a new and surprising mechanism that should be of significant interest to a wide range of readers. The paper is reasonably well-written. The information is presented logically, and the figures convey essential information. Citation of earlier relevant work is good. Some attention to grammar is needed, especially with regard to the use of articles. The paper claims that the absence of magnetic frustration in CoCrNi, relative to the five-component ‘parent’ alloy CoCrFeMnNi, produces a phase transformation from metastable fcc to the stable hcp phase upon the sequential passage of $1/6\langle 112 \rangle$ partial dislocations. Several scenarios are described in the

manuscript to illustrate this localized phase transformation. However, three points require additional analysis and discussion.”

Response: We thank the reviewer for a careful review of our manuscript.

Comment 1: “First, the slip-induced phase transformation involves the sequential passage of partial dislocations on every other $\langle 111 \rangle$ plane. The authors must explain why the system would choose to deform by the passage of partial dislocations on every OTHER $\langle 111 \rangle$ plane, rather than every $\langle 111 \rangle$ plane or $\langle 111 \rangle$ planes that are more widely separated. The authors clearly show that the total system energy is reduced for this situation AFTER the passage of the partial dislocations. However, how does the system ‘know’ this will be the result when a partial dislocation is first nucleated and before it propagates to form or grow the hcp phase? The possibility that the transformation occurs all at once, rather than growing progressively by the sequential nucleation and passage of many partial dislocations, also requires consideration and discussion.”

Response: We would like to draw the reviewer’s attention to Fig .5 (Fig. 4 in the original manuscript). The energetic pathway calculations not only show the energy of the system after passage of partials, but they also provide the barrier that needs to be overcome to reach that state. As Fig.5 (b) shows, the hcp formation is more favorable compared to esf formation with a slightly lower barrier. Subsequent growth of this hcp layer reduces the energy further (Path 1). In addition, another scenario is considered in Path 3, starting from an existing twin boundary. The competing processes are now hcp formation vs twin growth. As the calculations show, hcp formation both lowers the final energy and is associated with a smaller barrier. Similar arguments are typically made in the literature to explain the energetic pathways of twinning. While, this analysis clearly compares the energetics of hcp formation vs twin growth, we agree with the reviewer that it doesn’t provide a mechanistic picture of these processes. That is why we proceeded with the molecular dynamics simulations on a surrogate system to show various scenarios where hcp layers form as a result of interaction of lattice dislocations with internal boundaries. In other words, the analysis presented in Fig.5 first establishes the feasibility of hcp formation and then complementary simulations provide several mechanisms through which the hcp formation can actually occur.

Comment 2: “Second, the authors must discuss the trailing partials, which are not mentioned at all in the manuscript. Once the trailing partial travels through the material, the stacking of $\langle 111 \rangle$ planes is returned to the original sequence and the hcp phase is destroyed. The separation between leading and trailing

partials may be sufficiently small so that the growth of a thick hcp plate may be difficult. The authors must describe how the hcp phase may persist with respect to the movement of trailing partial dislocations.”

Response: We thank the reviewer for pointing out that we have not been sufficiently clear in presenting these results. The trailing partials are not neglected in any of the simulations. All simulations are started with introducing the perfect dislocation. In case of the mixed dislocation, we even explored the effect of having 90-degree and the 30-degree partials acting as both leading/trailing partials. We have tried numerous combinations of applied strain on various combinations of leading/trailing partials and only reported the combinations where hcp layers were formed with lowest level of strain possible.

In general, under zero strain condition, the two partials will repel one another without any obstacle due to the negative stacking fault energy, resulting in extended stacking faults. If there is enough resolved shear strain on the (111) planes the two partials can be forced to move in the same direction, but the extent of the stacking fault will depend on the strain and no equilibrium separation can be achieved.

In supplementary Figure S5, we show a combination of strain that results in a stress concentration at the leading partial, large enough to overcome the hcp formation barrier. Increasing strain beyond that point does not pass the dislocation through the boundary, instead, the partials that are traveling on the twin boundary will reach the fixed boundaries of the simulation, and nucleate more partials at the fixed boundaries, which keep the hcp layer growing. We did not include those snapshots, to avoid mixing artifacts of the fixed boundary conditions with actual process that result in the initial nucleation. We added the above explanation to the Supplementary materials, page 5, paragraph 3 under the section “Heterogeneous nucleation of hcp laths”.

In addition, the supplementary videos provide various interesting interactions of both 90 and 30 degree leading/trailing as well as partials corresponding to the screw dislocation with hcp layers.

However, we agree with the reviewer that under favorable loading conditions, other partials can effectively undo the hcp layers and revert the crystal back to fcc. Since the twin boundaries that acted as the nuclei for the initial hcp layer formation are still present, we envision that the hcp formation process can happen again with subsequent dislocation interactions on the intersecting slip system. This is part of the reason that hcp formation is beneficial for increasing strength without compromising plastic deformation. We have added more discussion at lines 262-269. We believe that the text in page 14, lines 247-269 address this concern.

Comment 3: “Finally, the authors do a good job of demonstrating that the fcc to hcp transition can occur by the passage of leading partial dislocations on selected $\langle 111 \rangle$ planes, but they do not adequately describe how this localized phase transformation produces strengthening. This new strengthening mechanism is the top-line discovery of this paper, and so the authors must convince the readers that this phase transformation is responsible for the additional strengthening observed in CoCrNi relative to CoCrFeMnNi. Showing that the passage of partial dislocations in CoCrFeMnNi does not produce a localized fcc to hcp transformation (see the additional comment below) could provide circumstantial support for the strengthening of CoCrNi relative to CoCrFeMnNi, but it would be more convincing to describe the specific mechanism by which this localized phase transformation strengthens CoCrNi.”

Response: All of the phenomena described in response to previous comment support our argument that the hcp formation provides an additional strengthening (it stops the gliding partials of mixed character traveling on the intersecting glide planes) without compromising plastic deformation: additional partials are then nucleated that either leave the hcp layers unchanged, grow the hcp layer, or unzip the hcp and revert it back to the perfect fcc structure depending on the sign of the nucleated partial.

Comment 4: “The authors demonstrate a localized fcc to hcp transformation associated with slip in CoCrNi by calculating the energy of the fcc and hcp phases (Fig. 3) and through simulations involving the sequential passage of partial dislocations (Fig. 4). Based on calculations of the energies of the fcc and hcp phases in CoCrFeMnNi in magnetic and non-magnetic states, the authors imply that slip does not induce this local phase transformation in CoCrFeMnNi. However, this is rather indirect proof of the absence of the localized fcc to hcp phase transformation, and it would be more convincing for the authors to perform simulations on CoCrFeMnNi as illustrated in Fig. 4 for CoCrNi. This is especially important since Fig. 3 shows that the hcp phase has a lower total energy than the fcc phase for some of the configurations in the magnetic state of CoCrFeMnNi (see Fig. 3b).”

Response: This point is addressed above in response to reviewer 2. In summary, we performed stacking fault calculations for the quinary alloy as well. These results are now shown in Fig 6 and the corresponding discussion is added to page 11, lines 188-198. Overall, the average behavior of both alloys follow similar trends; however, the deviation from average behavior is much more

significant in the quinary alloy. This is consistent with our previous predictions.

Comment 5.1: “Three additional, comments are provided to improve the quality, readability and accessibility of the work. First, Fig 1 will not be entirely clear to readers who are not expert electron microscopists. The term “center of symmetry” in the caption needs explanation in a way that’s accessible to the non-specialist, and the FCCM and FCCT labels on Figs. 1b,d require definition.”

Response: We thank the reviewer for this point, which was also raised by reviewer 1. We have addressed these concerns.

Comment 5.2: “Second, the work describes the strengthening of CoCrNi relative to CoCrFeMnNi. However, this could equivalently be considered as weakening of CoCrFeMnNi relative to CoCrNi and simpler alloys (binary alloys and/or alloys best represented as dilute solutions of elements). The authors should discuss this perspective to better support the claim of a new “strengthening” mechanism in the concentrated CoCrNi alloy. Specifically, if this same mechanism is found in concentrated and/or dilute binary alloys, then the high entropy alloy in the present paper is actually weaker than more conventional alloys. This is a subtle but important point, since much of the HEA literature proposes that HEAs will be stronger than alloys with fewer elements.”

Response: We appreciate the reviewer’s interesting take on this. We have added the following sentences to page 15 lines 276-278 “These findings emphasize the fact that “high entropy alloys” with more constituent elements are not necessary superior to their derivatives. Instead, the combination of chemical, and as emphasized presently, magnetic identities of elements govern the properties.”

Comment 5.3: “Finally, the order in which the elements are listed in the two alloys seems to be rather arbitrary (for example, NiCoCr rather than CoCrNi or CrCoNi). While there is no right or wrong order, this can make keyword searches less effective. The HEA field seems to be moving toward adoption of one of two standardized naming conventions – either listing the elements alphabetically or listing in order of increasing atomic number. It’s recommended that the authors consider using one of these two conventions.”

Response: We thank the reviewer for bringing this convention to our attention. We have adopted the increasing-atomic-number convention in naming the alloys.

REVIEWERS' COMMENTS:

Reviewer #1 (Remarks to the Author):

The reviewer is satisfied with responses from the authors. The new Fig.4, where they show why other ternary alloys do not have the driving force for the fcc to hcp phase transformation, is very helpful.

The reviewer now simply has another concern or more precisely, interest on the situations for quaternary alloys CoCrFeNi, CoFeNiMn, and CoCrNiMn, particularly CoCrFeNi. This is echoing what the authors claimed in the introduction part, where they said "the ease of twin formation alone cannot explain the drastic difference in behavior of this alloy compared to other twinning-deformation dominated materials, or the superiority of the ternary CrCoNi over the quinary CrMnFeCoNi". The issue here is that, if the driving force for the fcc to hcp phase transformation is also there for CoCrFeNi, how are they going to comment on the superiority of CoCrNi over CoCrFeNi? Or they can simply provide the evidence (similar to what they did in the new Fig. 4) that such a driving force is also missing for CoCrFeNi, then the problem is much easier.

Reviewer #2 (Remarks to the Author):

I have reviewed the authors' rebuttal letter and changes to the manuscript. To my mind, they have addressed my critique reasonably well, and I believe that I understand their argument much better than I did. Figure 4 makes their argument clearer that Mn is the most potent determining factor for the phase stability, and that Mn is also magnetically frustrated.

However, I am still not convinced that the frustration is the origin of the behavior. At first thought, I would not expect that frustration would alter the relative phase stability of FCC and HCP structures.

Both should be frustrated. Interestingly, the frustration of the Cr spins in NiCoCr does appear to depend on structure, with the FCC structure displaying stronger indications of frustration. However, in NiCoCr, the magnetism doesn't make much of a difference in the relative phase stability. In contrast, Mn does appear to be about equally frustrated in both structures of NiCoMn alloys, and yet, the effect of magnetism is to make energies of both structures comparable. So the correlation of the near degeneracy in formation energies with Mn and magnetism is certainly established, but the frustration aspect still seems puzzling to me. However, I expect that this is my shortcoming rather than the authors'.

Having noted that, I do think that the manuscript is now suitable for publication in Nature Communications. The authors have identified the HCP/FCC phase energy difference as a key factor for tuning the properties of these alloys, and have also noted that magnetism plays a large role in this relative phase stability. Both of these insights are very important and move the field forward.

Reviewer #3 (Remarks to the Author):

The authors have addressed all of my questions.

REVIEWERS' COMMENTS:

Reviewer #1 (Remarks to the Author):

Comment 1: “The reviewer is satisfied with responses from the authors. The new Fig.4, where they show why other ternary alloys do not have the driving force for the fcc to hcp phase transformation, is very helpful.”

Response: We are glad that we could clarify the issues raised by the reviewer previously and once again thank the reviewer for helpful suggestions.

Comment 2: “The reviewer now simply has another concern or more precisely, interest on the situations for quaternary alloys CoCrFeNi, CoFeNiMn, and CoCrNiMn, particularly CoCrFeNi. This is echoing what the authors claimed in the introduction part, where they said “the ease of twin formation alone cannot explain the drastic difference in behavior of this alloy compared to other twinning-deformation dominated materials, or the superiority of the ternary CrCoNi over the quinary CrMnFeCoNi”. The issue here is that, if the driving force for the fcc to hcp phase transformation is also there for CoCrFeNi, how are they going to comment on the superiority of CoCrNi over CoCrFeNi? Or they can simply provide the evidence (similar to what they did in the new Fig. 4) that such a driving force is also missing for CoCrFeNi, then the problem is much easier.”

Response: We agree with the reviewer that the case of quaternary derivatives is also interesting, however in order to make a conclusion more TEM and DFT analysis is required on this family of alloys. This is beyond the scope of the current manuscript which is centered around CrCoNi. The family of quaternary alloys should also be investigated and we plan to do so in subsequent studies.

Reviewer #2 (Remarks to the Author):

Comment 1: “I have reviewed the authors' rebuttal letter and changes to the manuscript. To my mind, they have addressed my critique reasonably well, and I believe that I understand their argument much better than I did. Figure 4 makes their argument clearer that Mn is the most potent determining factor for the phase stability, and that Mn is also magnetically frustrated.”

Response: We thank the reviewer again for raising several helpful issues in the first round of review. We are glad that we have been able to sufficiently address the concerns.

Comment 2: “However, I am still not convinced that the frustration is the origin of the behavior. At first thought, I would not expect that frustration would alter the relative phase stability of FCC and HCP structures. Both should be frustrated. Interestingly, the frustration of the Cr spins in NiCoCr does appear to depend on structure, with the FCC structure displaying stronger indications of frustration. However, in NiCoCr, the magnetism doesn't make much of a difference in the relative phase stability. In contrast, Mn does appear to be about equally frustrated in both structures of NiCoMn alloys, and yet, the effect of magnetism is to make energies of both structures comparable. So the correlation of the near degeneracy in formation energies with Mn and magnetism is certainly

established, but the frustration aspect still seems puzzling to me. However, I expect that this is my shortcoming rather than the authors'.

Response: We generally agree with the reviewer that our results establish a “correlation” between existence of magnetic frustration and comparable fcc/hcp energies but does not “prove causality”. We have reflected this in the response to the first round of reviews which is in agreement with the remarks above.

Comment 3: “Having noted that, I do think that the manuscript is now suitable for publication in Nature Communications. The authors have identified the HCP/FCC phase energy difference as a key factor for tuning the properties of these alloys, and have also noted that magnetism plays a large role in this relative phase stability. Both of these insights are very important and move the field forward.”

Response: We appreciate the reviewer’s thorough assessment of our manuscript.

Reviewer #3 (Remarks to the Author):

Comment 1: “The authors have addressed all of my questions.”

Response: We once again thank the reviewer for helpful suggestions during the first round of reviews. We are glad that the manuscript is now sufficiently clear.